# Extracted *Eucalyptus globulus* Bark Fiber as a Potential Substrate for *Pinus radiata* and *Quillaja saponaria* Germination

**DOI:** 10.3390/plants13060789

**Published:** 2024-03-11

**Authors:** Víctor Ferrer-Villasmil, Cecilia Fuentealba, Pablo Reyes-Contreras, Rafael Rubilar, Gustavo Cabrera-Barjas, Gastón Bravo-Arrepol, Danilo Escobar-Avello

**Affiliations:** 1Unidad de Desarrollo Tecnológico, Universidad de Concepción, Coronel 4191996, Chile; vjfv1975@gmail.com (V.F.-V.); gastonbravo@arrepol.com (G.B.-A.); 2Centro Nacional de Excelencia para la Industria de la Madera (CENAMAD), Pontificia Universidad Católica de Chile, Av. Vicuña Mackenna, 4860, Santiago 7820436, Chile; preyes@leitat.cl (P.R.-C.); rafaelrubilar@udec.cl (R.R.); 3Centro de Excelencia en Nanotecnología (CEN), LEITAT Chile, Santiago 7500618, Chile; 4Cooperativa de Productividad Forestal, Departamento de Silvicultura, Facultad de Ciencias Forestales, Universidad de Concepción, Concepción 4030000, Chile; 5Departamento de Silvicultura, Facultad de Ciencias Forestales, Universidad de Concepción, Concepción 4030000, Chile; 6Facultad de Ciencias para el Cuidado de la Salud, Universidad San Sebastián, Campus Las Tres Pascualas, Lientur 1457, Concepción 4080871, Chile; gustavo.cabrera@uss.cl; 7Facultad de Medicina y Ciencia, Universidad San Sebastián, Campus Las Tres Pascualas, Lientur 1457, Concepción 4080871, Chile

**Keywords:** sustainable agriculture, substrates, fibers, germination, forestry, waste management

## Abstract

This study aimed to explore alternative substrates for growing forest species using eucalyptus bark. It evaluated the potential of extracted *Eucalyptus globulus* fiber bark as a substitute for commercial growing media such as coconut fiber, moss, peat, and compost pine. We determined the physicochemical parameters of the growing media, the germination rate, and the mean fresh and dry weights of seedlings. We used the Munoo-Liisa Vitality Index (MLVI) test to evaluate the phytotoxicity of the bark alone and when mixed with commercial substrates. Generally, the best mixture for seed growth was 75% extracted eucalyptus bark fiber and 25% commercial substrates. In particular, the 75E-25P (peat) mixture is a promising substitute for seedling growth of *Pinus radiata*, achieving up to 3-times higher MLVI than the control peat alone. For *Quillaja saponaria*, the best growth substrate was the 50E-50C (coconut fiber) mixture, which had the most significant MLVI values (127%). We added chitosan and alginate-encapsulated fulvic acid phytostimulants to improve the performance of the substrate mixtures. The fulvic acid, encapsulated or not, significantly improved MLVI values in *Q. saponaria* species and *P. radiata* in concentrations between 0.05 and 0.1% *w*/*v*. This study suggests that mixtures with higher levels of extracted fiber are suitable for growing forest species, thus promoting the application of circular economy principles in forestry.

## 1. Introduction

*Eucalyptus* sp. is one of the most widely cultivated species worldwide, covering over 20 million hectares in Australia, Spain, Portugal, Kenya, Brazil, Uruguay, and Chile [1]. In Chile, commercial plantations of *E. globulus* and *E. nitens* covered 1.8 million hectares, and roundwood consumption was 13.64 million m^3^ of solid wood without bark in 2022 [2]. The harvesting of *Eucalyptus* sp. supports pulp production and, to a lesser extent, other manufactured products such as solid wood. One of the initial steps in processing eucalyptus logs is debarking, which generates significant amounts of bark as a by-product. Bark disposal and reuse present complex challenges that have become problematic in various countries and regions. According to Quihó et al. [3], eucalyptus bark production typically ranges from 8 to 10% of log volume. Chile produces 1.5 million cubic meters of eucalyptus bark each year, which is used for industrial boilers.

Eucalyptus bark residues have a complex morphology, low calorific value, and high ash content, making them unattractive for industrial fuel use and requiring blending with other biomass, such as pine bark and wood sawdust [4,5,6]. Research has focused on adding value to eucalyptus bark by evaluating its chemical composition and fiber potential, introducing solutions from a circular economy perspective. Eucalyptus bark can be a source of bioactive compounds with antioxidant, antifungal, and insecticidal properties [7]. It can be used to produce insulation boards [8,9], bioadhesives for particleboard manufacture [10,11], reinforcement in concrete [12], and growing substrates for horticulture and forestry [13,14,15,16]. Developing new alternative substrates for agricultural and nursery activities is crucial due to the need to incorporate more commercial options for replacing peat as a substrate medium [17]. Future challenges include the expensive cost of premium horticultural peat, particularly in nations lacking peat moss resources [18]. Therefore, it is relevant to find more sustainable alternatives. Peat could be replaced by organic waste or renewable materials like pine bark, sewage sludge, eucalyptus bark, biochar, husked rice, coco coir dust, and stabilized wood fiber for environmental benefits [16,19,20,21].

Coco fiber is often regarded as an excellent substrate due to its ability to be transformed into various particle sizes [14] and its favorable interaction with water. However, its limited availability has led many countries to import this raw material from places like India, Sri Lanka, and Vietnam, resulting in a negative environmental impact [22]. Forest and agricultural sector alternatives can reduce dependence on substrates such as peat and coco fiber [23]. As noted in several studies, there is an important opportunity to incorporate new renewable and sustainable organic materials into cropping systems in response to climate change. Substrate availability, sustainability, and performance metrics must be considered when proposing new substrate alternatives.

The main function of the substrate is to support plant growth [24]. Schafer et al. [25] mention that substrate quality depends on its physical structure and chemical composition. A good substrate must have several characteristics to ensure good plant growth. Substrates should have good nutrient and moisture retention capacity, good aeration, low resistance to root penetration, and good resistance to structural loss since they are used at a developmental stage when plants are susceptible to attack by microorganisms and less tolerant to water deficit [17]. The main difficulties with substrates are their ability to maintain the proper moisture level, a sufficient supply of nutrients, and aeration. All these aspects directly impact seedling germination, development, and final plant quality [26].

For over 50 years, bark has been used as a horticultural medium. Still, it has been deemed unsuitable due to its insufficient C/N ratio and the presence of phytotoxins like phenolic, triterpene, and tannic compounds. These compounds can inhibit plant development when used as a growing medium. Various approaches have been studied to reduce or eliminate the phytotoxicity of bark as growing media. In this sense, Guedes et al. [13] and Buamscha et al. [27] have all shown that treatments improve germination and radicle growth with less inhibitory effects compared to fresh bark biomass.

Eucalyptus bark has been studied as an organic growing media for horticultural applications by Chemetova et al. [14] and Escobar-Avello et al. [16]. Chemetova et al. [14] have suggested that eucalyptus bark can be treated at low temperatures in a hydrothermal process to eliminate its phytotoxic effect. However, this treatment should be used in a mixture with peat, not more than 25% by volume, to improve aeration and maintain good water content compared to commercial substrates. Escobar-Avello et al. [16] have considered reducing the phytotoxicity of eucalyptus bark by conducting a water extraction process and mixing eucalyptus and pine bark during extraction. Their results showed that eucalyptus bark treated at 70 °C for 70 min can replace up to 75% of commercial substrates such as coconut fiber, peat moss, and peat, for the seed germination and growth of horticultural species, such as radish (*Raphanus sativus*) and Chinese cabbage (*Brassica rapa*). However, although water-extracted *E. globulus* bark has previously been studied as a growth medium for germinating horticultural crops, to the best of our knowledge, its use for germinating forest tree seeds such as *P. radiata* and *Q. saponaria* has not been reported.

A substrate’s physical characteristics are crucial for plant growth, and they should interact positively with added plant nutrients. Adding a plant biostimulant to the substrate could enhance seed germination and further the growth and development of plantlets [28]. Plant biostimulants include well-known products such as chitosan, humic and fulvic acid (humic substances), and protein hydrolysates, among others [29,30]. Chitosan is a natural polysaccharide formed chemically by glucosamine units β-(1-4) linked. It can be obtained by deacetylation of chitin from crustacean shells, fungal mycelium, or directly from fungal biomass [31,32]. Humic substances are a mixture of organic substances that improve soil quality and facilitate plant nutrient uptake [33]. They are produced through the decomposition of plant and animal residues, catalyzed by environmental factors and microorganisms [34]. On the other hand, agrochemical encapsulation is a well-known strategy that protects active ingredients from environmental degradation or leaching, and controls and prolongs their release [35,36]. Encapsulated biostimulants can be a valuable tool for promoting and maintaining healthy plant development when added to substrates [37,38].

Our study aimed to investigate alternative substrates based on eucalyptus bark for use as nursery growing media for forest species. We chose two species, the first being *P. radiata* D. Don, due to its great commercial importance in Chile, and the second being *Q. saponaria*, a native fast-growing species that is usually chosen as an option for the reforestation of eroded, fire-ravaged soils or restoration purposes. The high volume of traditional substrates used in forest nurseries makes replacing some or all of them with forest by-products of low commercial value and high availability attractive.

## 2. Results and Discussion

### 2.1. Chemical Properties of the Substrates

The chemical composition of *E. globulus* fiber bark and the extracted *E. globulus* fiber as growing media is presented in Table 1. A complete physicochemical characterization of the produced growing media has been reported previously by Escobar-Avello et al. [16]. These properties are critical to understanding the characteristics and suitability of each substrate, whether for use as forestry, agricultural, or horticultural substrates. Although water-extracted *E. globulus* bark has previously been used as a growth medium for germinating crops [14,15,16,39], to the best of our knowledge, its use for germinating forest tree seeds such as *P. radiata* and *Q. saponaria* has not been reported.

The pH value plays a crucial role in regulating the ion exchange equilibrium between the nutrient reserves, the soil colloids, and the soil solution [40]. The pH values showed a slight decrease with the pre-treatment of the bark water extraction, shifting from 6.0 to 5.5 (a reduction of 8.3%) but remaining within the recommended range for organic substrates of nursery species. Remarkably, it is possible to observe that *P. radiata* stands to grow over a wide range of site conditions, from a low pH of 4.67 to a neutral pH of 6.18 [41]. On the other hand, there was a sharp decrease in the electrical conductivity values, from 316 µS/cm in the eucalyptus bark before extraction to 124 µS/cm in the pre-treated bark (a decrease of 60.7%). This result suggests that soluble salts and nutrients were probably leached out during the aqueous extraction of the bark [14]. More importantly, pH and conductivity levels suggest no salt problem effects on the seedling or cutting development of exotic or native species in the eucalyptus-extracted growing media [42,43].

All substrates had high organic matter (OM) and carbon contents, and no relevant differences in these parameters were observed after treatment (0.9% in both cases). High OM substrates provide adequate porous media for root growth and the water-holding capacity required for plant growth and development between irrigation events [44].

The bark of eucalyptus fiber has a low nitrogen content, with approximately 50% being removed during heat treatment. The composition of the carbon source, which includes lignin and cellulose, may affect nutrient immobilization. A higher C/N ratio reduces organic matter decomposition, preserving soil structure, but may minimize nitrogen availability if not appropriately managed in the growing media via fertilization [45]. However, our observed C/N ratio for both eucalyptus substrates (<242:1) was lower than that of the boiled sawdust (500:1) used successfully by Santelices and Bobadilla [46] as rooting growing media for *Q. saponaria* cuttings when fertilized and disinfected periodically.

After treatment, the moisture content of the eucalyptus bark fiber decreases while the ash content remains low. Lower moisture content makes substrates easier to store and handle. The ash content affects the electrical conductivity of the substrate. On the other hand, the NH_4_/NO_3_ ratio obtained from eucalyptus bark fiber was high (4.3). However, this value decreased after treatment. The availability of NH_4_ and NO_3_ can affect plant growth and development, with NH_4_ being more readily available to plants than NO_3_ [47]. After the extraction process, the levels of various macronutrients and micronutrients in the substrates, including P_2_O_5_, K_2_O, CaO, MgO, and Na, decrease considerably. The decrease in the growing media’s nutritional content is highly desirable in large-scale seedling or cutting production forest nursery processes as growing media nutrient availability may be better controlled via fertigation or direct fertilization [43].

### 2.2. Substrate Physical Properties

Both extracted and non-extracted eucalyptus fiber bark have a particle size distribution concentrated at more than 50% in the size range of <2 mm (37%) and 2.0–4.0 mm (28%) (Table 2). Understanding the physical characteristics and management of substrates, particularly in terms of aeration and water retention capacity, is pertinent to the variation in particle size. Ideal growing media with a balanced particle size ranging from 0.8 mm to 6 mm secures good root growth, adequate plant water use, and excess water drainage after irrigation for conifer species [48]. Quiroz et al. [49] have suggested that particle sizes between 20 to 40% lower than 0.8 mm are appropriate for growing native Chilean species using composted pine bark.

The bulk density of eucalyptus substrates and extracted eucalyptus fiber bark is notably low, at around 0.03 g/mL. These prepared substrates meet the bulk density requirements outlined in the Chilean Standard 2880 (<0.7 g/mL) [50]. However, these eucalyptus substrates seem very low compared to the 0.1 to 0.45 g/mL bulk densities recommended by Quiroz et al. [49] as effective growing media. Nevertheless, the lower bulk density of eucalyptus substrates will not limit root growth. Still, retaining water may be less efficient, and more frequent irrigation may be required for growing plants, or a combination with other growing medias may be required for optimizing tree nursery operations [51]. These results agree with the SEM observations (Figure 1). Additionally, whether extracted or not, eucalyptus fiber exhibits the highest water-holding capacity, with a slight increase observed after the extraction process. 

#### Surface Characteristics of Eucalyptus Fiber Barks

Figure 1 shows the results of the SEM analysis of the eucalyptus fiber bark before and after treatment.

Initially, the surface of raw eucalyptus fiber bark is rough and fibrous due to a dense structure where individual fibers are closely packed together. However, treatment leads to greater disruption of the cell wall, resulting in a higher fiber exposure compared to the control. This transformation causes the surface to become highly porous, with observable broken cells, which can be attributed to the solubilization of cell wall components such as cellulose, hemicellulose, and lignin. These results confirm the effectiveness of the treatment in modifying the surface properties and morphology of eucalyptus bark. The findings are consistent with our previous research, which revealed a tissue of broken cells covered with residues resulting from the extraction process [52].

### 2.3. Pine and Quillay Substrate Germination Rates

Seed germination rate (GR) is a critical factor in plant growth and development and is influenced by various factors such as temperature, water supply, mineral nutrition, and light [53]. The crucial ecological and economic significance lies in the seed’s capacity to germinate and initiate plant growth at the appropriate seasonal timing [54]. This work evaluated the germination speed of *Q. saponaria* and *P. radiata* seeds using extracted and unextracted eucalyptus fiber mixed with commercial substrates (coconut fiber, moss, peat, and composted pine) as growing media.

In general, the germination rate for *Q. saponaria* was higher than for *P. radiata*. The extraction process appeared to favor germination for both species. However, in this study, no significant differences were observed. Previous research by Chemetova et al. [15] demonstrates that hydrothermal treatments successfully removed phytotoxicity from *E. globulus* fresh bark. In addition, phytotoxic elements in fresh bark-based growing media have been reported to affect plant development [14,55]. The maximum GR for *P. radiata* was recorded in substrates that used extracted eucalyptus fiber (50E-50C and 75E-25P) and commercial CP (Table 3).

No significant difference was observed in using extracted eucalyptus fiber in *Q. saponaria* species, whether the fiber was treated or not. In other cases, a higher GR was recorded for seeds sown on unextracted eucalyptus fiber or commercial substrates (75E-25C, C, M, CP), while in some other cases, the GR was higher when the fiber was extracted (25E-75M, 50E-50CP).

Considering that the low GR may be due to phytotoxic compounds, the results do not explain the behavior of all evaluated substrates. For example, in the case of *Q. saponaria*, commercial substrates of coconut fiber, moss, and composted pine promoted a 100% GR, which was affected by whether there was incorporation in the mixture of extracted eucalyptus or not. The peat mix is noteworthy since the maximum GR was obtained under all the evaluated combinations, which contrasts with the 67% GR observed when only peat was used as a substrate; therefore, a positive effect was observed in the mixtures.

The germination rate for *P. radiata* was low, even compared to commercial substrates, and in a few mixtures (50E-50C and 75E-25P), the positive effect previously observed for *Q. saponaria* seeds was recorded. The results suggest differences in the media conditions that favor the germination of both seeds. It has been reported that *Q. saponaria* seeds are highly affected by soil matric potential, with northern seed sources performing better under low soil matric potential [56]. For *Pinus pinea* seeds, it was observed that the germination is influenced by drought, salinity, and heat, with the highest germination percentage presented at 80 °C and short exposure time [57]. The best performance being of composted pine growing substrate can be explained by its high water-holding capacity and potential for improving nutrient availability [13,58].

Table 4 shows the mean weight (fresh and dry) of germinated plants per pot for the evaluated mixtures. The positive effect of using the extracted eucalyptus fiber on plant biomass is highlighted, specifically for 25E-75C and 50E-50CP mixtures evaluated for *Q. saponaria* media where higher mean fresh plant weight and better plant growth were observed. These mixtures suggest that any phytotoxic compounds were diminished or removed. Statistical analyses indicate no relevant difference between commercial substrates for most mixtures in NEB or EEB, supporting the idea that they could be used as a substitute for commercial growing media. For instance, in the case of *Q. saponaria* for EEB, the 25E-75C duplicates the mean weight compared to the moss substrate control treatment (102.8 vs. 48.6 mg pot^−1^). For *P. radiata* seedlings, there was no apparent effect of using the extracted eucalyptus fiber, highlighting the high values observed in the fresh weight for some mixtures with NEB (75E-25C, 25E-75C, 75E-25M, 75E-25P). When EEB was used, mixtures with peat, specifically 75E-25P and 50E-50P, showed the highest mean values for plant weight. The observed responses for the different mixtures suggest species-specific results regarding growth in extracted and non-extracted eucalyptus growing media fiber.

The best performances of the extracted eucalyptus fiber mixtures with moss and peat may be due to improved water retention and aeration properties [15,59]. For the two species studied, extracted eucalyptus fiber promoted plant growth with higher fresh weight (74–94%) compared to non-extracted eucalyptus fiber (53–94%). Therefore, water availability is favored for substrates with extracted eucalyptus fiber.

The mixtures’ mean dry weights indicate the biomass to compare among all tested combinations. However, it is important to consider that germination was not the same for all samples, so weights should be compared along similar germination percentages. Statistical analysis for *Q. saponaria* seedling dry weights germinated in NEB and EEB indicates no significant differences among treatment means, including the commercial substrate (moss). The same trend was observed for *P. radiata* seedlings. Our results suggest that mixtures with eucalyptus bark by-products with regular commercial substrates may be a suitable alternative growing media.

### 2.4. Evaluation of Phytostimulants on the Growth of Q. saponaria and P. radiata Species

Phytostimulants are increasingly used in sustainable agriculture because they enhance plant growth and promote tolerance to biotic and abiotic stresses [60,61]. These natural compounds can directly provide plants with mineral nutrients and influence their hormone biosynthesis and homeostasis, resulting in improved growth and development [62]. In horticultural production, phytostimulants have been shown to promote plant growth, increase stress tolerance, and enhance fruit quality [63]. Using phytostimulants from natural origins offers a more eco-friendly and sustainable approach to plant growth and crop production.

This study evaluated the effect of using phytostimulants such as chitosan and fulvic acid in mixtures containing 75% of extracted eucalyptus fiber and 25% commercial substrates on the growth of *Q. saponaria* and *P. radiata* species. This mixture was chosen because it generally gives the highest total fresh sample weights. The selected phytostimulants are valuable additives due to their diverse biological activities and potential applications in agriculture [64,65,66,67,68].

#### 2.4.1. Germination Rate for *Q. saponaria* and *P. radiata* Using Phytostimulants

Table 5 shows the germination rates of the species studied for the different mixtures and under the different concentrations of evaluated phytostimulants (water was included as a control).

The species *Q. saponaria* showed the highest germination rates, especially for mixtures containing coconut fiber and composted pine (75E-25C and 75E-25CP). The effect of chitosan on this species is striking, as germination was zero in the 75E-25M mixture at any concentration used. The analysis of the pH of this solution showed that it was acidic (pH = 5), which could have affected germination. For *P. radiata*, germination rates remained low, most of the evaluated treatments between 33 and 67%, and no relevant treatment effect was observed due to the use of phytostimulants.

#### 2.4.2. Effect of Phyto-Stimulant on *Q. saponaria* and *P. radiata* Seedling Biomass

Table 6 shows the fresh and dry mean weight of *Q. saponaria* seedlings growing on EEB and commercial substrates mixtures. The statistical analysis indicates that there were no important differences among treatments, including water as a control. This result suggests that the evaluated substrates and treatments behave similarly to commercial substrates. In this case, it was impossible to establish an optimal phytostimulant concentration or any encapsulation effect. However, encapsulation of this additive has many advantages, such as protection of the active agent, reduction in the amount of phytostimulant, and controlled release of the encapsulated molecule [69,70]. The amount of moisture in the seedlings growing in each mixture was higher than 80%, similar to that reported in the tests without phytostimulant (Table 4) and on substrates with extracted eucalyptus fiber. However, the mean dry weight of the mixture did not change significantly. This result suggests an increase in biomass as an effect of the phytostimulant.

In the case of *P. radiata* (see Table 7), the substrates 75E-25M and 75E-25P showed an increase in mean fresh plant weight compared to the water control, while for the 75E-25C (FAe3) and 75E-25CP (FAue1, FAe1) substrates, a similar mean fresh weight was obtained only in specific cases. This observed behavior contrasts with *Q. saponaria*, where no significant differences were reported for all substrates (Table 6). This outcome implies that the efficacy of specific phytostimulants for certain plant species may be due to their ability to induce particular defense responses and enhance the synthesis of secondary metabolites [71,72]. The effect of encapsulation was most relevant for the 75E-25M mixture, with an optimum concentration of 0.1%. In contrast, for the 75E-25P substrate, the highest mean fresh weight was recorded for the non-encapsulated fulvic acid at a concentration of 0.05%. The moisture content of the seedlings growing on the different mixtures follows the trend observed for *Q. saponaria* plants, with a moisture content of over 80%. The increase in biomass because of the phytostimulant is more pronounced in this species, with 55–220% increases in the mean dry weight compared to the control in specific cases (75E-25M and 75E-25P).

Various factors, including cost, availability, and environmental sustainability, drive the need for alternative substrates in plant cultivation. Peat, a commonly used substrate, is becoming more expensive and difficult to obtain [73]. Due to the increasing demand and cost of peat and its unpredictable availability because of environmental restrictions, the search for alternative high-quality and cost-effective nursery growing media materials in horticulture is essential [74]. Our study evaluated a widely available forest by-product (i.e., eucalyptus bark) as a possible full or partial replacement for commercial growing media such as coconut fiber, moss, peat, and composted pine. Successful *Q. saponaria* and *P. radiata* growth tests from seeds were performed on extracted eucalyptus fiber bark substrates mixed with commercial substrates at different ratios (25, 50, and 75% *v*/*v*).

### 2.5. Phytotoxicity of Substrate Mixtures Measured by Munoo-Liisa Vitality Index (MLVI)

#### Eucalyptus/Commercial Substrate Mixtures Growing *Q. saponaria* and *P. radiata*

The MLVI values for *Q. saponaria* on substrates prepared with extracted and non-extracted eucalyptus mixes and commercial substrates is shown in Figure 2. As previously stated, raw eucalyptus fiber bark (NEB) might harm MLVI values. EEB alone was an excellent growth medium, with MLVI values in the same range as those observed for commercial substrates such as moss and coconut fiber. The same result was observed when EEB was added in a mixture with the commercial substrates, giving even better outcomes than these. Compared to its pure equivalent (coconut fiber MLVI = 100%, control) and the remaining commercial substrates, the 50E-50C mix had the most significant MLVI values (127%). The other combinations followed the same pattern. The case of peat stands out, of which the MLVI value practically doubled with the incorporation of EEB, going from 50% in NEB to 108% in the 25E-75P mixture. This synergistic substrate effect on plant growth is critical for farmers and nursery businesses worldwide. It also justifies searching for the best substrate combination to create a germinating and growing material. The same behavior has been reported previously in other research carried out by our group in radish (*Raphanus sativus*) growth trials on mixtures of commercial substrates and extracted eucalyptus fiber [16].

Our results show that EEB can act as a substitute with a significant percentage depending on the commercial substrate. For example, EEB can replace 25–50% of CP for composted pine, giving similar or even better results than CP alone (80% vs. 100% MLVI). In the case of peat, any mixture promotes an improvement in MLVI, with the best being the 25E-75P mixture. For moss, an improvement in MLVI was observed with increasing EEB replacement (MLVI = 127%), while for coconut fiber, the best mixture was 50E-50C (MLVI = 123%). This result is positive because it could reduce the use of peat, which is no longer considered a renewable resource due to the long time it takes to regenerate [75]. The better performance of the substrate mix could be explained by the presence of the favorable porosity conditions and water-holding capacity provided by the EEB. (See Table 2). In this sense, the work by Chemetova et al. [14] has shown that hydrothermally treated peat mixtures with eucalyptus bark provide substrates with good aeration properties.

Figure 3 shows the MLVI values for *P. radiata* obtained in the different eucalyptus fiber barks in a mixture with commercial substrates.

The difference between using EEB instead of NEB is widely noted. EEB alone was shown to be superior to the commercial substrates tested. EEB gave the highest MLVI values for most mixtures, which were well above those obtained with the commercial substrates (peat used as control). The case of the 75E-25P mix is striking, as the MLVI obtained was up to 3-times higher than that of control peat alone (300% vs. 100% MLVI). Once again, the synergy mentioned above was observed in the *Q. saponaria* trials. The excellent performances of other mixtures, such as 75E-25CP, 25E-75P, 25E-75M, and 50E-50C, is also noteworthy. The optimal amount of EEB replacement for the growth of the *P. radiata* species may vary depending on the commercial substrate used. Beyond the optimal mix, it is evident that EEB can become a good substitute for commercial substrates. We suggest that this excellent performance can be explained by combining several factors in the peat-EEB mixture, such as good aeration, organic matter content, nitrogen, and phosphorus availability, and a high proportion of nitrogen and phosphorus [76].

### 2.6. Phytotoxicity of Substrate Mixtures with Phytostimulants Measured by Munoo-Liisa Vitality Index (MLVI)

Phytostimulant tests were conducted on the mixtures with the highest commercial substrate replacement (75% EEB). As previously mentioned, phytostimulants are crucial in sustainable agriculture by promoting plant growth and enhancing stress tolerance [60,61], providing nutrients to plants, and influencing phytohormone action [62]. The encapsulation of phytostimulants can improve their delivery and efficacy [60]. This technology allows for the production of customized micro- and nanoparticles made of a coating material containing a primary active ingredient [77]. For this reason, in this research, we used fulvic acid as a phytostimulant encapsulated in an alginate matrix. We also included unencapsulated fulvic acid, chitosan (a commercial biostimulant), and water for comparison.

Figure 4 and Figure 5 show the MLVI values for *Q. saponaria* and *P. radiata* growth using phytostimulants (peat was used as a control for MLVI determination in both cases.). For the species *Q. saponaria*, a significant improvement in its MLVI values was observed with the presence of fulvic acid (encapsulated or not), except for the 75E-25M mixture, which recorded its highest MLVI with the control. In general, the optimum concentration of fulvic acid remained between 0.05 and 0.1%, and whether it was encapsulated seems irrelevant. In the case of the 75E-25C mixture, encapsulation of the biostimulant for the 0.05% concentration had a positive effect compared to its unencapsulated counterpart. A significant decrease in MLVI was recorded when chitosan was used, except for the 75E-25CP mixture. As discussed above (Table 5), the acidic pH of this additive (close to 5) may have affected seedling growth. For the substrate containing composted pine (75E-25CP), we have a slightly basic pH (~7.7), suggesting a hypothesis that the combination of the chitosan solution with this substrate creates more favorable pH conditions. We observed the same behavior as when a trial used radish and Chinese cabbage [16].

For *P. radiata*, a similar behavior to that previously discussed for *Q. saponaria* was observed when the different phytostimulants were used. The best growth substrate was the 75E-25C mixture, which increased the MLVI value by up to 40% (compared to the control) when unencapsulated fulvic acid was used at a concentration of 0.5%. In this case, the pH of the chitosan solution was not as relevant as observed for *Q. saponaria*, and the growth of *P. radiata* seedlings was recorded for all the mixtures evaluated, even with MLVI values higher than those obtained with the control. In this sense, some researchers have pointed out that the application of chitosan to different pine species enhanced growth parameters and improved seedling quality and nutrient utilization [78,79]. Our result shows that the nature of the species is also an essential factor to be considered.

The MLVI and germination results showed that a mixture of EEB and commercial substrates such as peat, coconut fiber, or composted pine with a higher percentage of extracted fiber (75%) could be used to cultivate these forest species.

## 3. Materials and Methods

### 3.1. Collection and Preparation of Substrates

Forestal Collicura (Santa Juana, Bio-Bio region, Chile) provided eucalyptus (*E. globulus*) bark. In addition, the eucalyptus bark was screened to a size of 20 mm to separate sticks and chips from the bark. The bark was then ground in a hammer mill (Breuer model M8, BTD, St. Vith, Belgium). The fiber obtained was sieved to a 4 mm size to remove dust and small stones. The resulting fiber was subsequently named “eucalyptus fiber bark” (E). In the germination tests, commercial substrates commonly used for these purposes were used for comparison. These included coconut fiber (C), peat moss (M), composted pine (CP), and peat (P).

### 3.2. Pilot-Scale Extraction of Eucalyptus Bark

The extraction of eucalyptus bark was carried out in a 25 L steel reactor with a stirrer under optimal conditions (temperature: 70 °C, time: 70 min, substrate to water ratio 1:10, *w*/*w*), as reported in previous research [16]. The reactor was heated by an electric resistance with a temperature controller.

### 3.3. Substrate Physicochemical Characteristics

#### 3.3.1. pH Measurement

The samples’ pH values were determined according to the standard method of UNE-EN 13037 [80]. Briefly, the pH of the solution after sedimentation was measured using a PL-700PC pH/conductivity meter (Gondo, Taipei, Taiwan) by mixing 300 mL of water at 20 °C with 60 mL of the material and stirring for 1 h.

#### 3.3.2. Determination of Electrical Conductivity

Electrical conductivity was measured using a PL-700PC pH/conductivity meter, according to the standard method described in the UNE-EN 13038 standard [81]. The same procedure was used to measure the pH, but the final evaluated solution was filtered after stirring.

#### 3.3.3. Organics and Ashes Determination

Organic matter and ash determinations were carried out following the standard method of UNE-EN 13039 [82]. First, in a capsule that had been previously calcined and weighed, a 5 g substrate sample was dried at 105 °C for 4 h. After drying, the capsule was allowed to cool in a desiccator and weighed. Dried sample capsules were placed in a muffle and calcined at 450 °C for 6 h. Following calcination, capsules were allowed to cool in a desiccator before being weighed again. Sample moisture, organic matter, and ash contents were then calculated using the Test Methods for the Examination of Composting and Compost (TMECC) 05.07 and 04.02 [83,84].

#### 3.3.4. Bulk Density Determination

Bulk density was determined using the standard method TMECC standard 03.03 [85]. First, the weight of an empty 2000 mL beaker was recorded. Then, an aliquot of the sample was transferred to a 600 mL beaker to fill the 2000 mL beaker. This procedure was repeated two more times. Each time, the sample was allowed to fall freely from a height of 15 cm until the beaker was filled with 1800 mL of fluid (the third time, the sample was not allowed to fall freely from a height of 15 cm).

#### 3.3.5. N-NO_3_, N-NH_4_, and Chemical Elements Determination

Nutrient availability considering N-NO_3_, N-NH_4_, P_2_O_5_, K_2_O, CaO, MgO, and Na was evaluated using the standard method described by the TMECC [84,86,87]. The nitrate content in the samples was determined via UV absorption at 220 nm and 275 nm according to the nitrate ion method (TMECC 04.02-B). The colorimetric phenol hypochlorite composting method, described in the Standard Methods [84], was used to determine ammonium nitrogen in the compost beds. In this procedure, NH_3_ was determined spectrophotometrically at 635 nm by forming the strong indophenol blue compound through the reaction of NH_3_ with HClO and phenol.

#### 3.3.6. Scanning Electron Microscopy (SEM)

The morphological properties and surface characteristics of eucalyptus fiber bark before and after treatment were determined by using scanning electron microscopy (SEM) utilizing a JEOL JSM-6380 microscope (Tokyo, Japan). The operating voltage of the microscope was 20 kV acceleration voltage. The samples were dried at room temperature and then coated with a conductive gold layer of about 150 Å thickness using a sputtering apparatus Edwards S 150 (Agar Scientific, Standsted, UK) [88].

### 3.4. Phytostimulants

#### 3.4.1. Fulvic Acid Encapsulation

The ionic gelation method used sodium alginate (Sigma-Aldrich, reagent grade, St. Louis, MO, USA) as the encapsulation matrix of commercial fulvic acid (phytostimulant). First, a 2% (*w*/*v*) fulvic acid solution in Milli-Q (Merck, Rahway, NJ, USA) water was mixed with a 1.75% (*w*/*v*) sodium alginate solution with stirring for 15 min at 40 °C to increase the solubility of the alginate. The resulting solution was homogenized in an Ultra Turrax IKA^®^ T25 digital (Staufen, Germany) at 10,000 rpm twice for 1 min (with a 1 min break) and filled into a syringe. This solution was then dropped into a beaker containing an 8% CaCl_2_ (*w*/*v*) solution at room temperature and agitated continuously at 200 rpm, waiting for the drops to gel as they fell into the beaker. The alginate–fulvic acid microspheres were immediately filtered and washed with distilled water as soon as they were formed. Finally, the gel spheres were kept dry to prevent the diffusion of fulvic acid.

#### 3.4.2. Preparation of Chitosan

Crab chitosan (degree of deacetylation 86%, Mw 65 kDa) was previously prepared in our laboratory [31]. A solution of the biopolymer (1 wt%) was prepared by dissolving an appropriate amount (1 g) in 0.175 M acetic acid and stirring at 60 °C until a homogeneous solution was obtained. Before use, the pH was adjusted to 5 with a 1 M NaOH solution. Different concentrations of chitosan (0.05, 0.1, and 0.5 wt%) were prepared from this solution. Each chitosan solution containing the corresponding substrate received an equal addition of alginate-encapsulated fulvic acid (dry weight base).

### 3.5. Growth Evaluations

#### 3.5.1. Phytotoxicity Essay

According to the European standards UNE-EN 16086-2 [89], a total of three *P. radiata* and *Q. saponaria* seeds were incubated in a plastic pot filled (by triplicate, n = 3) with a substrate (60 cm^3^) at room temperature (25 °C) in the dark for 6 days. Peat commercial substrate was used as control (P). The Munoo-Liisa vitality index (MLVI; Equation (1)) was used to evaluate the phytotoxicity considering the germination rate (GR) and root length (RL).
(1)$$Munno\text{-}Liisa\;Vitality\;Index\left(\%\right)=\frac{\left({GR}_{1}\cdot {RL}_{1}+{GR}_{2}\cdot {RL}_{2}+{GR}_{3}\cdot {RL}_{3}\right)}{3\cdot {GR}_{P}\cdot {RL}_{P}}\cdot 100$$
where RL_1–3_ and GR_1–3_ are triplicates, and RL_P_ and GR_P_ correspond to the control. To calculate the MLVI parameter, the commercial substrate that promotes greater root length and germination was used as a control.

#### 3.5.2. Growth Test

The UNE-EN 16086-2 Spanish standard was modified to evaluate the growth test [89]. First, three seeds of *Q. saponaria* or *P. radiata* were sown in a container of 150 cm^3^. Then, leaf soil was placed at the bottom of the container, and 60 mL of the substrate containing the seed was placed on top. The substrate consisted of 25:75, 50:50, and 75:25 (by volume) mixtures of eucalyptus fiber (extracted and unextracted) with commercial peat, coconut fiber, moss, or composted pine substrates. The commercial substrates were also included as controls. The experiments were conducted at 21 °C for 21 days for *Q. Saponaria,* and 31 days for *P. radiata*. The fresh weight of the seedlings was then measured, and the dry weight was determined by placing the seedlings in an oven at 65 °C for 48 h. The mixtures of 75% extracted eucalyptus fiber and 25% commercial substrates (coconut, moss, peat, and composted pine) were used in the phytostimulant trials.

### 3.6. Statistical Analysis

Data were analyzed through a 2-way analysis of variance as well as Tukey’s and Holm–Sidak multiple comparison tests. Statistics were conducted using Prism 8 for Windows 10 (Graphpad software, version 8.0, San Diego, CA, USA). *p*-values below 0.05 were considered significant for all tests and mean comparisons.

## 4. Conclusions

The study concluded that *P. radiata* and *Q. saponaria* seedlings grew in substrate mixtures containing 75% bark fiber and 25% commercial media. The addition of eucalyptus substrate mixed with other commercial substrates further improved the performance of the substrate, as evidenced by the MLVI, germination data, and mean fresh and dry weights. The 75E-25P peat mixture is a promising substitute for seedling growth of the *P. radiata* species, achieving up to 3-times higher MLVI than the control peat, while the 50E-50C coconut fiber mixture is the best for *Q. saponaria*. Additionally, fulvic acid phytostimulants enhanced the performance of the substrate even more. This study suggests that fulvic acid use (0.05–0.1% *w*/*v*), both encapsulated and unencapsulated, can improve the Munoo-Liisa Vitality Index (MLVI) values for the *P. radiata* and *Q. saponaria* species. This finding supports the implementation of circular economy concepts in forestry by utilizing combinations with a higher percentage of extracted eucalyptus fiber for growing forest species.

## Figures and Tables

**Figure 1 plants-13-00789-f001:**
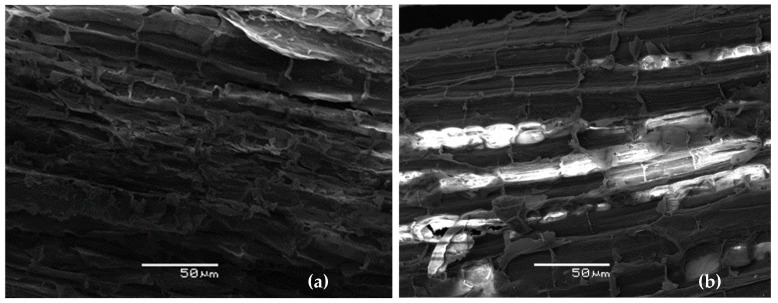
SEM images of raw (**a**) eucalyptus fiber bark and (**b**) extracted eucalyptus fiber.

**Figure 2 plants-13-00789-f002:**
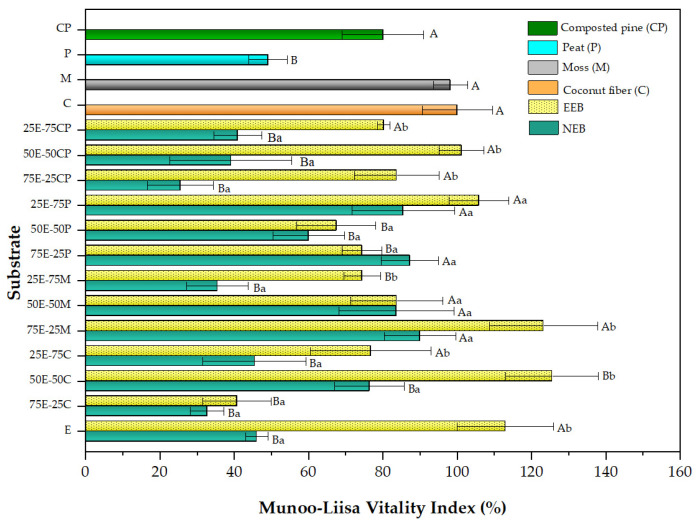
Munoo-Liisa Index for *Q. saponaria* in substrates prepared from extracted (EEB) and non-extracted (NEB) eucalyptus with mixtures of coconut fiber (C), moss (M), peat (P), and composted pine (CP). The capital letters indicate a significant difference compared to coconut fiber, and the lowercase letters indicate a significant difference between the extracted and non-extracted samples (two-way ANOVA Holm–Sidak’s multiple comparisons test, with α = 0.05).

**Figure 3 plants-13-00789-f003:**
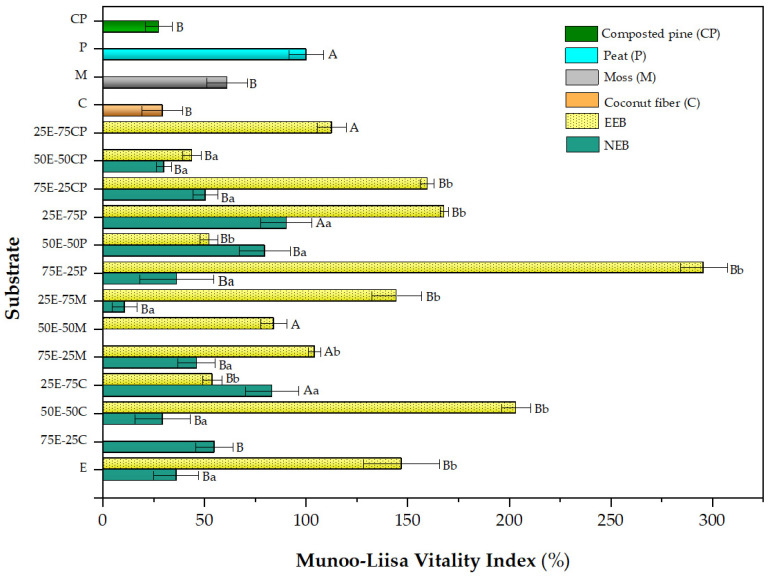
Munoo-Liisa Index for Pine in substrates prepared from extracted (EEB) and non-extracted (NEB) eucalyptus fiber bark with mixtures of coconut fiber (C), moss (M), peat (P), and composted pine (CP). The capital letters indicate a significant difference compared to peat, and the lowercase letters indicate a significant difference between the extracted and non-extracted samples (two-way ANOVA Holm–Sidak’s multiple comparisons test, with α = 0.05).

**Figure 4 plants-13-00789-f004:**
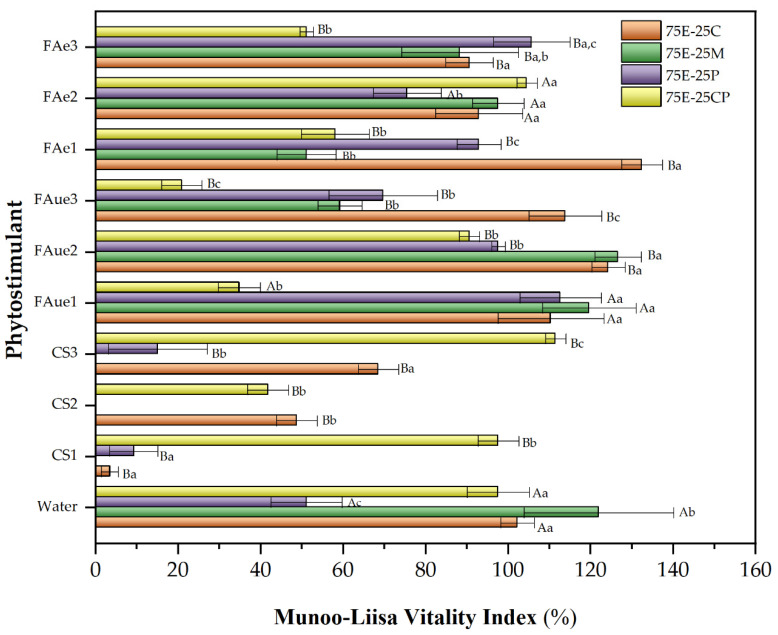
Munoo-Liisa Index for *Q. saponaria* in substrates prepared from extracted eucalyptus fiber bark (EEB) with coconut fiber (C), moss (M), peat (P), and composted pine (CP) mixtures, using phytostimulants (CS: chitosan; FA_ue_: unencapsulated fulvic acid; FA_e_: encapsulated fulvic acid). Numbers 1, 2, or 3 correspond to the concentration of phytostimulant: 1 = 0.05% *w*/*v*; 2 = 0.1% *w*/*v*; 3 = 0.5% *w*/*v*). The capital letters indicate a significant difference between treatments compared to water (two-way ANOVA Holm–Sidak’s multiple comparisons tests, with α = 0.05). The lowercase letters indicate a significant difference between samples under the same treatment (two-way ANOVA Tukey’s multiple comparisons tests, with α = 0.05).

**Figure 5 plants-13-00789-f005:**
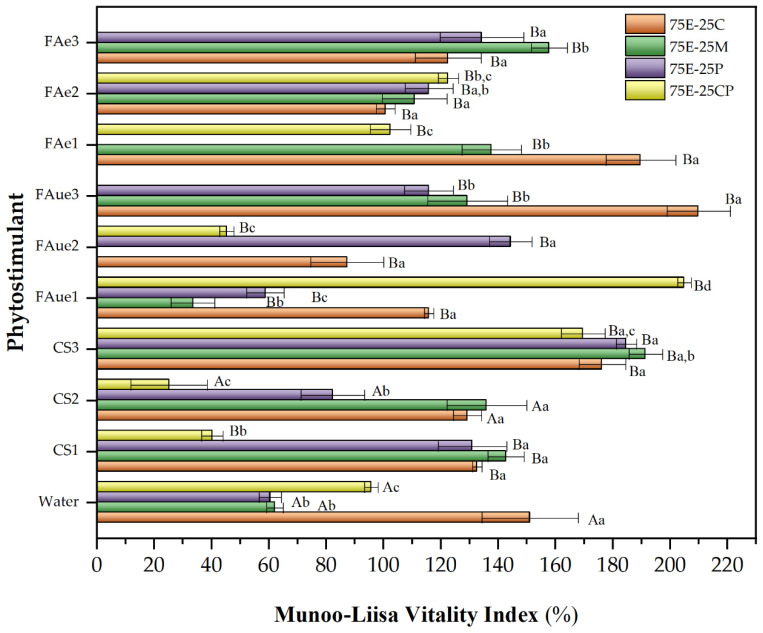
Munoo-Liisa Index for *P. radiata* in substrates prepared from extracted eucalyptus fiber bark (EEB) with coconut fiber (C), moss (M), peat (P), and composted pine (CP) mixtures, using phytostimulants (CS: chitosan; FAue: unencapsulated fulvic acid; FAe: encapsulated fulvic acid. Numbers 1, 2, or 3 correspond to the concentration of phytostimulant: 1 = 0.05% *w*/*v*; 2 = 0.1% *w*/*v*; 3 = 0.5% *w*/*v*). The capital letters indicate a significant difference between treatments compared to water (two-way ANOVA Holm–Sidak’s multiple comparisons tests, with α = 0.05). The lowercase letters indicate a significant difference between samples under the same treatment (two-way ANOVA Tukey’s multiple comparisons tests, with α = 0.05).

**Table 1 plants-13-00789-t001:** Physicochemical properties of substrates eucalyptus fiber and extracted eucalyptus fiber.

Category	Property	Raw Material (%)	Extracted Fiber (%)	Difference (%)
Chemical Composition	pH	6.0	5.5	−8.3%
	Electrical Conductivity (μS/cm)	316.0	124.0	−60.7%
	Organic Matter	93.7	94.5	0.9%
	Organic Carbon	52.8	53.3	0.9%
	Total Nitrogen	0.4	0.2	−48.8%
	C/N Ratio	123.0	242.0	96.7%
Moisture	Humidity (%)	11.1	6.6	−40.3%
Ash Content	Ash (%)	6.3	5.5	−13.0%
Nitrogen Fractions (mg/Kg)	N-NH_4_	574.0	119.0	−79.3%
	N-NO_3_	133.0	112.0	−15.8%
	NH_4_/NO_3_ Ratio	4.3	1.1	−74.4%
Minerals (%)	P_2_O_5_	0.2	0.1	−50.0%
	K_2_O	0.4	0.1	−68.4%
	CaO	1.4	1.4	0.7%
	MgO	0.3	0.2	−31.2%

The extraction process for the *E. globulus* fiber bark was carried out at 70 °C for 70 min. Adapted from Escobar-Avello et al. [16]. Reproduced with permission from Escobar-Avello et al., Pretreated *Eucalyptus globulus* and Barks: Potential Substrates to Improve Seed Germination for a Sustainable Horticulture; published by Forests, 2023.

**Table 2 plants-13-00789-t002:** Presents a comparative analysis of the main physical properties of substrates between eucalyptus fiber bark in its raw state and the bark after extraction.

Properties		Particle Size (mm)			Bulk Density (g/mL)	Pore Space (%)	Free Air Space (%)	WRC (%*v*/*v*)
		<2	2–4	4–16				
Raw material	Eucalyptus fiber bark	37	28	35	0.030	98	39	59
	Eucalyptus fiber bark extracted	37	29	34	0.028	97	32	65

Adapted from Escobar-Avello et al. [16]. Reproduced with permission from Escobar-Avello et al., [16] Pretreated *Eucalyptus globulus* and *Pinus radiata* Barks: Potential Substrates to Improve Seed Germination for a Sustainable Horticulture; published by Forests, 2023.

**Table 3 plants-13-00789-t003:** Germination rate for *Q. saponaria* and *P. radiata* in substrates prepared from extracted (EEB) and non-extracted (NEB) eucalyptus fiber bark with coconut fiber (C), moss (M), peat (P), and composted pine (CP) mixtures.

Mixture	Germination Rate, GR (%)			
	*Q. saponaria*		*P. radiata*	
	NEB	EEB	NEB	EEB
E	100 ^Aa^	100 ^Aa^	67 ^Aa^	67 ^Aa^
75E-25C	67 ^Aa^	33 ^Aa^	33 ^Aa^	0 ^Aa^
50E-50C	100 ^Aa^	100 ^Aa^	33 ^Aa^	100 ^Aa^
25E-75C	100 ^Aa^	67 ^Aa^	67 ^Aa^	33 ^Aa^
75E-25M	100 ^Aa^	100 ^Aa^	33 ^Aa^	33 ^Aa^
50E-50M	100 ^Aa^	100 ^Aa^	0 ^Aa^	33 ^Aa^
25E-75M	67 ^Aa^	100 ^Aa^	33 ^Aa^	67 ^Aa^
75E-25P	100 ^Aa^	100 ^Aa^	33 ^Aa^	100 ^Aa^
50E-50P	100 ^Aa^	100 ^Aa^	67 ^Aa^	33 ^Aa^
25E-75P	100 ^Aa^	100 ^Aa^	67 ^Aa^	67 ^Aa^
75E-25CP	100 ^Aa^	100 ^Aa^	67 ^Aa^	67 ^Aa^
50E-50CP	67 ^Aa^	100 ^Aa^	33 ^Aa^	33 ^Aa^
25E-75CP	67 ^Aa^	67 ^Aa^	0 ^Aa^	67 ^Aa^
C	100 ^A^		33 ^A^	
M	100 ^A^		67 ^A^	
P	67 ^A^		67 ^A^	
CP	100 ^A^		100 ^A^	

NEB: non-extracted eucalyptus fiber bark, EEB: extracted eucalyptus fiber bark, C: coconut fiber, M: moss, P: peat, CP: composted pine. The capital letters on the side of each mean indicate significant differences compared to the control treatment mean for each species (composted pine for both *Q. saponaria* and *P. radiata*). Small lowercase letters indicate a significant difference between mixtures of the same commercial substrates for each species (one-way ANOVA Holm–Sidak’s multiple comparisons test, with α = 0.05).

**Table 4 plants-13-00789-t004:** Fresh and dry mean weight of *Q. saponaria* and *P. radiata* seedlings growing in eucalyptus fiber bark and commercial substrates mixtures.

Mixture	Mean Fresh Weight (mg pot^−1^)			Mean Dry Weight (mg pot^−1^)	
	*Q. saponaria*	*P. radiata*		*Q. saponaria*	*P. radiata*
			NEB		
NEB	18.0 ^A^	97.9 ^A^		8.4 ^A^	30.5 ^A^
75E-25C	30.2 ^Aa^	170.8 ^Ba^		6.9 ^Aa^	18.2 ^Ba^
50E-50C	45.4 ^Aa^	109.0 ^Ab^		7.7 ^Aa^	11.3 ^Ba^
25E-75C	27.7 ^Aa^	226.1 ^Bc^		8.0 ^Aa^	23.0 ^Aa^
75E-25M	43.3 ^Aa^	200.7 ^Ba^		6.3 ^Aa^	36.0 ^Aa^
50E-50M	43.8 ^Aa^	0 ^Bb^		7.7 ^Aa^	0 ^Bb^
25E-75M	41.7 ^Aa^	57.7 ^Bc^		7.0 ^Aa^	14.9 ^Bc^
75E-25P	40.9 ^Aa^	197.6 ^Ba^		6.5 ^Aa^	10.8 ^Ba^
50E-50P	34.6 ^Aa^	138.7 ^Ab^		8.1 ^Aa^	24.9 ^Aa^
25E-75P	39.8 ^Aa^	114.9 ^Ab^		9.3 ^Aa^	24.5 ^Aa^
75E-25CP	16.3 ^Aa^	115.1 ^Aa^		10.1 ^Aa^	30.5 ^Aa^
50E-50CP	34.6 ^Aa^	139.4 ^Aa^		5.3 ^Aa^	24.4 ^Ab^
25E-75CP	34.2 ^Aa^	0 ^Bb^		6.1 ^Aa^	0 ^Bc^
C	35.1 ^A^	111.8 ^A^		7.5 ^A^	43.6 ^B^
M	48.6 ^A^	107.3 ^A^		8.3 ^A^	51.9 ^B^
P	31.1 ^A^	121.9 ^A^		6.9 ^A^	32.1 ^A^
CP	32.9 ^A^	82.9 ^A^		9.5 ^A^	35.6 ^A^
			EEB		
EEB	46.5 ^A^	160.1 ^A^		6.7 ^A^	12.1 ^B^
75E-25C	55.5 ^Aa^	0 ^Ba^		7.4 ^Aa^	0 ^Ba^
50E-50C	46.2 ^Ab^	167.9 ^Ab^		8.0 ^Aa^	21.3 ^Ab^
25E-75C	102.8 ^Bc^	152.7 ^Ab^		5.9 ^Aa^	22.2 ^Ab^
75E-25M	51.8 ^Aa^	170.8 ^Aa^		5.6 ^Aa^	32.2 ^Aa^
50E-50M	49.5 ^Aa^	155.2 ^Aa^		5.0 ^Aa^	39.5 ^Ab^
25E-75M	32.7 ^Aa^	145.7 ^Aa^		5.4 ^Aa^	15.6 ^Bc^
75E-25P	50.7 ^Aa^	178.2 ^Ba^		5.2 ^Aa^	32.9 ^Aa^
50E-50P	51.5 ^Aa^	175.3 ^Ba^		6.0 ^Aa^	11.7 ^Bb^
25E-75P	46.2 ^Aa^	152.7 ^Aa^		6.2 ^Aa^	28.7 ^Aa^
75E-25CP	48.5 ^Aa^	168.8 ^Aa^		6.5 ^Aa^	21.0 ^Aa^
50E-50CP	67.7 ^Ba^	169.9 ^Aa^		11.8 ^Aa^	34.8 ^Ab^
25E-75CP	55.5 ^A^	135.3 ^Ba^		6.3 ^Aa^	16.6 ^Ba^

NEB: non-extracted eucalyptus fiber bark, EEB: extracted eucalyptus fiber bark, C: coconut fiber, M: moss, P: peat, CP: composted pine. The capital letters on the side of each mean indicate significant differences compared to the control treatment mean for each species (moss for *Q. saponaria* and peat for *P. radiata*). Small lowercase letters on the side of each mean indicate a significant difference between mixtures of the same commercial substrates for each species (one-way ANOVA Holm–Sidak’s multiple comparisons test, with α = 0.05).

**Table 5 plants-13-00789-t005:** Germination rate of *Q. saponaria* and *P. radiata* using phytostimulants in substrate mixtures prepared from extracted (EEB) eucalyptus fiber bark with coconut fiber (C), moss (M), peat (P), and composted pine bark (CP) mixtures.

Sample	Germination Rate (%)							
	75E-25C		75E-25M		75E-25P		75E-25CP	
	*Q. saponaria*	*P. radiata*	*Q. saponaria*	*P. radiata*	*Q. saponaria*	*P. radiata*	*Q. saponaria*	*P. radiata*
Water	100 ^A^	67 ^A^	100 ^A^	33 ^A^	33 ^A^	33 ^A^	100 ^A^	67 ^A^
CS_1_	33 ^A^	67 ^A^	0 ^B^	67 ^A^	67 ^A^	67 ^A^	100 ^A^	33 ^A^
CS_2_	67 ^A^	67 ^A^	0 ^B^	67 ^A^	0 ^A,^	33 ^A^	100 ^A^	33 ^A^
CS_3_	67 ^A^	100 ^A^	0 ^B^	67 ^A^	67 ^A^	67 ^A^	100 ^A^	100 ^A^
FA_ue1_	100 ^A^	67 ^A^	100 ^A^	33 ^A^	100 ^A^	33 ^A^	100 ^A^	67 ^A^
FA_ue2_	100 ^A^	33 ^A^	100 ^A^	0 ^A^	67 ^A^	33 ^A^	100 ^A^	33 ^A^
FA_ue3_	100 ^A^	67 ^A^	100 ^A^	67 ^A^	67 ^A^	33 ^A^	100 ^A^	0 ^A^
FA_e1_	100 ^A^	67 ^A^	67 ^A^	33 ^A^	100 ^A^	0 ^A^	67 ^A^	67 ^A^
FA_e2_	100 ^A^	67 ^A^	100 ^A^	33 ^A^	67 ^A^	67 ^A^	100 ^A^	100 ^A^
FA_e3_	100 ^A^	67 ^A^	100 ^A^	67 ^A^	100 ^A^	67 ^A^	67 ^A^	0 ^A^

CS: chitosan; FA_ue_: unencapsulated fulvic acid; FA_e_: encapsulated fulvic acid. Numbers 1, 2, or 3 corresponds to the concentration of phytostimulant: 1 = 0.05% *w*/*v*; 2 = 0.1% *w*/*v*; 3 = 0.5% *w*/*v*. The capital letters on the side of each mean indicate significant differences compared to the control treatment mean for each species water for both *Q. saponaria* and *P*. *radiata* (one-way ANOVA Holm–Sidak’s multiple comparisons test, with α = 0.05).

**Table 6 plants-13-00789-t006:** Fresh and dry mean weights of germinated *Q. saponaria* seedlings growing on mixtures of extracted eucalyptus fiber bark and commercial substrates.

Phytostimulant	Fresh and Dry Mean Weight (mg pot^−1^)							
	75E-25C		75E-25M		75E-25P		75E-25CP	
	Fresh	Dry	Fresh	Dry	Fresh	Dry	Fresh	Dry
Water	42.5 ^A^	4.9 ^A^	52.9 ^A^	8.1 ^A^	65.7 ^A^	8.2 ^A^	37.8 ^A^	7.7 ^A^
CS_1_	29.0 ^Aa^	5.4 ^Aa^	0 ^B^	0 ^B^	26.9 ^Ba^	6.4 ^Aa^	36.8 ^Aa^	4.9 ^Aa^
CS_2_	35.4 ^Aa^	5.6 ^Aa^	0 ^B^	0 ^B^	0 ^Bb^	0 ^Bb^	35.8 ^Aa^	5.1 ^Aa^
CS_3_	46.9 ^Aa^	7.2 ^Aa^	0 ^B^	0 ^B^	25.1 ^Ba^	6.9 ^Aa^	48.1 ^Aa^	6.3 ^Aa^
FA_ue1_	53.0 ^Aa^	7.9 ^Aa^	58.4 ^Aa^	7.7 ^Aa^	61.7 ^Aa^	8.1 ^Aa^	43.2 ^Aa^	6.2 ^Aa^
FA_ue2_	50.3 ^Aa^	6.1 ^Aa^	57.7 ^Aa^	6.9 ^Aa^	48.5 ^Aa^	7.0 ^Aa^	45.9 ^Aa^	7.8 ^Aa^
FA_ue3_	56.4 ^Aa^	6.3 ^Aa^	31.9 ^Bb^	6.1 ^Aa^	50.4 ^Aa^	8.9 ^Aa^	29.0 ^Aa^	7.0 ^Aa^
FA_e1_	54.7 ^Aa^	7.2 ^Aa^	43.5 ^Aa^	7.9 ^Aa^	47.4 ^Aa^	7.4 ^Aa^	49.1 ^Aa^	6.9 ^Aa^
FA_e2_	34.8 ^Aa^	5.7 ^Aa^	44.1 ^Aa^	5.5 ^Aa^	49.2 ^Aa^	6.6 ^Aa^	47.8 ^Aa^	6.7 ^Aa^
FA_e3_	36.6 ^Aa^	5.6 ^Aa^	47.5 ^Aa^	5.8 ^Aa^	52.8 ^Aa^	6.9 ^Aa^	45.6 ^Aa^	7.0 ^Aa^

CS: chitosan; FA_ue_: unencapsulated fulvic acid; FA_e_: encapsulated fulvic acid. Numbers 1, 2, or 3 correspond to the concentration of phytostimulant: 1 = 0.05% *w*/*v*; 2 = 0.1% *w*/*v*; 3 = 0.5% *w*/*v*. The capital letters indicate a significant difference compared to water, and the lowercase letters indicate a significant difference between the mix of the same commercial substrates (two-way ANOVA Holm–Sidak’s multiple comparisons test, with α = 0.05).

**Table 7 plants-13-00789-t007:** Fresh and dry mean weights of germinated *P. radiata* seedlings growing on mixtures of extracted eucalyptus fiber bark and commercial substrates.

Phytostimulant	Fresh and Dry Mean Weight (mg pot^−1^)							
	75E-25C		75E-25M		75E-25P		75E-25CP	
	Fresh	Dry	Fresh	Dry	Fresh	Dry	Fresh	Dry
Water	205.2 ^A^	33.9 ^A^	128.0 ^A^	14.5 ^A^	92.9 ^A^	11.4 ^A^	170.3 ^A^	18.5 ^A^
CS_1_	141.1 ^Ba^	15.7 ^Ba^	154.8 ^Ba^	15.6 ^Aa^	129.6 ^Ba^	13.8 ^Aa^	91.7 ^Ba^	15.7 ^Aa^
CS_2_	142.1 ^Ba^	15.8 ^Ba^	182.5 ^Bb^	18.2 ^Aa^	149.4 ^Ba^	16.8 ^Aab^	109.3 ^Ba^	19.1 ^Aa^
CS_3_	175.3 ^Bb^	19.2 ^Ba^	183.0 ^Bb^	20.1 ^Aa^	191.4 ^Bb^	23.6 ^Bb^	107.8 ^Ba^	12.4 ^Aa^
FA_ue1_	172.9 ^Ba^	18.7 ^Ba^	92.7 ^Ba^	12.8 ^Aa^	238.3 ^Ba^	25.3 ^Ba^	170.2 ^Aa^	20.3 ^Aa^
FA_ue2_	138.1 ^Bb^	14.6 ^Ba^	0 ^Bb^	0 ^Bb^	181.1 ^Bb^	19.7 ^Ba^	142.8 ^Bb^	20.1 ^Aa^
FA_ue3_	150.4 ^Bab^	17.5 ^Ba^	160.3 ^Bc^	19.5 ^Ac^	216.7 ^Ba^	24.8 ^Ba^	0 ^Bc^	0 ^Bb^
FA_e1_	167.9 ^Ba^	17.3 ^Ba^	150.8 ^Aa^	16.4 ^Aa^	0 ^Ba^	0 ^Ba^	157.6 ^Aa^	18.4 ^Aa^
FA_e2_	159.9 ^Ba^	37.8 ^Ab^	210.5 ^Bb^	22.6 ^Bab^	154.1 ^Bb^	16.3 ^Ab^	123.5 ^Bb^	18.1 ^Aa^
FA_e3_	209.4 ^Ab^	23.5 ^Bac^	171.2 ^Ba^	28.5 ^Bb^	165.7 ^Bb^	17.2 ^Ab^	0 ^Bc^	0 ^Bb^

CS: chitosan; FA_ue_: unencapsulated fulvic acid; FA_e_: encapsulated fulvic acid. Numbers 1, 2, or 3 correspond to the concentration of phytostimulant: 1 = 0.05% *w*/*v*; 2 = 0.1% *w*/*v*; 3 = 0.5% *w*/*v*. The capital letters indicate a significant difference compared to water, and the lowercase letters indicate a significant difference between the mix of the same commercial substrates (two-way ANOVA Holm–Sidak’s multiple comparisons test, with α = 0.05).

## Data Availability

Data are contained within the article.

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
