# Peer review of "Extracted Eucalyptus globulus Bark Fiber as a Potential Substrate for Pinus radiata and Quillaja saponaria Germination"

_plants, 2024, doi:10.3390/plants13060789_

Round 1

Reviewer 1 Report

Comments and Suggestions for Authors

The manuscript entitled “Extracted Eucalyptus globulus bark as a potential substrate for Pinus radiata and Quillaja saponaria germination” contains information interesting of both scientific and practical point of view. Therefore I would recommend its publication after addressing several issues:

- Major concerns. Being a continuation of the work already published in the Forests journal (reference [16] in the work) it is important unless a special permission has been obtained, not to repeat (even partially) graphical representations and Tables constructions, which have been already been published in the other Journal. Even though in the legend of Table 1, for example, it has been stated that “**These results were previously reported in the study by Escobar-Avello et al. [16].”, this is not enough. IN ADDITION TO citing the published work as "previously reported" it is obligatory to modify its graphical representation. The same remark applied to  Table 2 and Figure 1.

- In addition some minor concerns need to also be addressed:

Abstract - please mention the whole genus names of species upon first mentioning

Introduction

- Abbreviation “swb” needs to be explained upon first mentioning

- Lines 67-68:  Please, elaborate on the essence of this claim and whether the cited reference discusses the issue in the point of view which it is cited here.

- Line 83: it needs to be “mention”

- Line 94: it needs to be “triterpene”

Please discuss on how this works represents a broadening of the research of the one conducted in reference [16].

Results and discussion

As already mentioned pay special attention to not repeat graphically already published work, even though the source has been mentioned.

Table 3 and Table 5 - Insert statistical processing of the data

Author Response

Responses to reviewer 1.

  • The manuscript entitled “Extracted Eucalyptus globulusbark as a potential substrate for Pinus radiata and Quillaja saponaria germination” contains information interesting of both scientific and practical point of view. Therefore, I would recommend its publication after addressing several issues:

  • Thank you for your feedback and interest in our research. We're glad to hear that you recommend our results for publication.

  • Major concerns. Being a continuation of the work already published in the Forests journal (reference [16] in the work) it is important unless a special permission has been obtained, not to repeat (even partially) graphical representations and Tables constructions, which have been already been published in the other Journal. Even though in the legend of Table 1, for example, it has been stated that “**These results were previously reported in the study by Escobar-Avello et al. [16].”, this is not enough. IN ADDITION TO citing the published work as "previously reported" it is obligatory to modify its graphical representation. The same remark applied to  Table 2 and Figure 1.

  • Thank you for your comment and concern. We have carefully reviewed the copyright policy of MDPI (https://www.mdpi.com/authors/rights). We understand that "For all articles published in MDPI journals, copyright remains with the authors". The original article Escobar-Avello et al [16] corresponds to the same research team of the present work. Both were led by Danilo Escobar-Avello. As mentioned in the page "Permission is not required for reconstruction of your own table with data already published elsewhere". As far as we know, we are not violating any copyrights.

However, we are including Tables 1 and 2 in the Supplementary Material so that previously published information is not duplicated in the main text. We believe that the information on the physicochemical properties of the substrates should be available in each article independently so that readers can consult this information directly in each article.

The case of Figure 1 is different since it corresponds to an SEM image of the fiber used specifically for this work. In addition, it is at a different magnification than the previous work.

In the footer of each table in the Supplementary Material, we use exactly the format followed by MDPI, so that it now reads: "Adapted from Escobar-Avello et al [16]. Reproduced with permission from Escobar-Avello et al. Pretreated Eucalyptus globulus and Barks: Potential Substrates to Improve Seed Germination for a Sustainable Horticulture; published by Forests, 2023"

In addition some minor concerns need to also be addressed:

  • Abstract - please mention the whole genus names of species upon first mentioning
  • Thank you for your comment. We appreciate your suggestion and have made the necessary changes to the species genera as you suggested.

Introduction

  • Abbreviation “swb” needs to be explained upon first mentioning
  • Thank you for your comment. The abbreviation “swb” was replaced by the description "solid wood without bark, " which expresses the same thing.

  • Lines 67-68:  Please, elaborate on the essence of this claim and whether the cited reference discusses the issue in the point of view which it is cited here.

  • Thank you for your comment. We have revised the sentence according to your advice and the context of the text.

  • Line 83: it needs to be “mention”

  • Thank you for your comment. The word has been fixed.

  • Line 94: it needs to be “triterpene”

  • Thank you for your comment. The word has been fixed.

  • Please discuss on how this works represents a broadening of the research of the one conducted in reference [16].
  • Thank you for your inquiry. Our previous publication demonstrated a high degree of species specificity in substrate utilization, which is also highlighted in our current research on forest species. It is important to note that the germination requirements for forest seedlings differ significantly from those of horticultural species. Our articles aim to establish a customized substrate for each species.

This study focuses on two important forest species in Chile. Our research aims to provide valuable insights into the germination dynamics of Pinus radiata, the most common forest species in the country, and Quillaja saponaria, a native species with commercial and ecological restoration significance. Our research aims to provide valuable insights into the germination dynamics of Pinus radiata, the most common forest species in the country, and Quillaja saponaria, a native species with commercial and ecological restoration significance, expanding upon the commercial application (on horticulture) referenced in the study [16].

In the introduction, we highlighted the contrast between the articles by rephrasing the following phrase: "However, although water-extracted E. globulus bark has previously been studied as a growth medium for germinating horticultural crops, to the best of our knowledge, its use for germinating forest tree seeds such as P. radiata and Q. saponaria has not been reported."

Results and discussion

  • As already mentioned pay special attention to not repeat graphically already published work, even though the source has been mentioned.

  • Thank you for your comment. Your concern was addressed in a previous response.

  • Table 3 and Table 5 - Insert statistical processing of the data

Thank you for your comment. We have included statistical processing in both tables and slightly modified the text to align with statistical aspects

Reviewer 2 Report

Comments and Suggestions for Authors

The paper includes two experiments on the use of Eucalyptus bark fiber on Quillaja saponaria and Pinus radiata seedlings. The first one examined the fiber as a medium component, the other presented the results of biostimulant treatment on seeddlings. The paper is well prepared, the aims, results and discussion are clearly stated and logically explained. The subject is actual, responding to the requirements for sustainable horticulture/nursery production. The language is properly used  and makes the article easy to read and understand. The authors treated the subject thoroughly, analyzed the Eucalyptus fiber from different sides (germination, biomass of seedling, SEM pictures of the fiber, phytotoxity) as well as biostimulants such as chitosan and fulvic acid. Yet, I give some remarks, which could improve the paper:

1.      In the title a word “fibre’ should be added. Otherwise the title is misleading.

2.      I think that the word “extracted’ is not so accurate and may mislead the reader. It will be better to use “processed” .

3.      If it is possible it would be valid to analyze the content of phytotoxic ingredients in fiber.

4.      In Table 3 the unextracted fiber is noted with letters UEB in the title, but EEB in the table.

5.      What’s the difference between peat and peat moss?

6.      The description of the procedure of SEM should be improved.

7.      A few further remarks are given in the text.

Author Response

Responses to reviewer 2.

The paper includes two experiments on the use of Eucalyptus bark fiber on Quillaja saponaria and Pinus radiata seedlings. The first one examined the fiber as a medium component, the other presented the results of biostimulant treatment on seeddlings. The paper is well prepared, the aims, results and discussion are clearly stated and logically explained. The subject is actual, responding to the requirements for sustainable horticulture/nursery production. The language is properly used  and makes the article easy to read and understand. The authors treated the subject thoroughly, analyzed the Eucalyptus fiber from different sides (germination, biomass of seedling, SEM pictures of the fiber, phytotoxity) as well as biostimulants such as chitosan and fulvic acid. Yet, I give some remarks, which could improve the paper:

1.In the title a word “fibre’ should be added. Otherwise the title is misleading.

  • Thank you for your comment. We include the word in the title accordingly.

2.I think that the word “extracted’ is not so accurate and may mislead the reader. It will be better to use “processed” .

  • Thank you for your suggestions. We believe that the word "extracted" better reflects what was intended to be done in this research, as it seeks to remove phytotoxic and/or non-beneficial compounds through a previous aqueous extraction step, as described in the article by Escobar-Avello (2023) (DOI: 10.3390/f14050991).

3.If it is possible it would be valid to analyze the content of phytotoxic ingredients in fiber.

  • Thank you for your suggestions. Analyzing the phytotoxic components present in eucalyptus fiber is an extensive and laborious task that could extend the scope of this research. However, it has been previously reported that eucalyptus fiber extracts are rich in compounds such as proanthocyanidins, which are also known as condensed tannins. It is also possible that Fisetinidol, Quercetin, Gallic acid, and isorhamnetin are present in high concentrations, among others compound more detailled are reported in previous work of Santos et al. (2023) (DOI: 10.3390/f14050895).

4.In Table 3 the unextracted fiber is noted with letters UEB in the title, but EEB in the table.

  • Thank you for your feedback. We apologize for the mistake and have made the necessary correction. The correct abbreviation is NEB: Non-extracted eucalyptus fiber bark.

5.What’s the difference between peat and peat moss?

  • Thank you for your comment. Although their names are similar, peat and peat moss are different materials with some of the same characteristics. Peat is a brown material with a composition similar to that of soil and usually grows in acidic, boggy soils. Peat is composed of decaying plant matter and enriches the area to enhance plant life when added to the soil. Peat moss can consist of sapric, hemic, or fibric materials, each with its level of decomposition. Peat moss consists of an absorbent moss that increases the acidity level of soils when added to it. Peat moss helps retain water in the soil, adds a body to sandy soils, and loosens soils containing a large amount of heavy clay.6. The description of the procedure of SEM should be improved.

6.The description of the procedure of SEM should be improved.

  • Thank you for your comment. More details on both the methodology and results were incorporated.

7.A few further remarks are given in the text.

  • Thank you for your comment. We have made the correction according to your suggestion.

Round 2

Reviewer 1 Report

Comments and Suggestions for Authors

Authors have addressed some of the issues with some of them still needing to be addressed.

- Regarding the issue of one and the same graphical repetition of data in authors’ publication.

As also mentioned by the authors "Permission is not required for reconstruction of your own table with data already published elsewhere". This was exactly what was requested in the first review. To modify, to graphically change, reconstruct the representation of the Tables, but not repeat them in the exactly same way which they were. So the Tables could remain also in the main text. But they must  be graphically changed (or “reconstructed” as written in the MDPI guidelines). Even if the Tables are now moved to the Supplementary they need to be graphically changed. In the guidelines (https://www.mdpi.com/authors/rights) it has been instructed:

“It is absolutely essential that authors obtain permission to reproduce any published material (figures, schemes, tables or any extract of a text) which does not fall into the public domain, or for which they do not hold the copyright. Permission should be requested by the authors from the copyright holder (usually the Publisher, please refer to the imprint of the individual publications to identify the copyright holder).

Permission is required for:

- Use of Tables, Graphs, Charts, Schemes and Artworks if they are unaltered or slightly modified.”

So, no matter if the Tables remain in the main text or go to the Supplementary, please modify their appearance. Although both publications are from one and the same MDPI publisher, it is strongly recommended to alter the graphical representation as a matter of good taste, the content being a part of a scientific publication.

- regarding statistical processing in Table 3: please insert the statistical processing on the values, similarly as it has been done in the other Tables.

Author Response

Thank you for your feedback. We appreciate your guidance on the graphical presentation of Tables 1 and 2. In response to your suggestion, we have modified the graphical representation of both tables and incorporated the specified footnote. We have also relocated these tables into the main article.

Regarding the statistical processing in Table 3, we have included the required statistical analysis on the values, aligning it with the methodology used in the other tables.

We sincerely hope that with these modifications you will find our article worthy of publication.

Round 3

Reviewer 1 Report

Comments and Suggestions for Authors

The authors have addressed the issues of the first and second reviewes and now the manuscript can be published as it is.